# Bacterial Communities Associated with the Leaves and the Roots of Salt Marsh Plants of Bayfront Beach, Mobile, Alabama, USA

**DOI:** 10.3390/microorganisms12081595

**Published:** 2024-08-06

**Authors:** Aqsa Majeed, Jinbao Liu, Adelle J. Knight, Karolina M. Pajerowska-Mukhtar, M. Shahid Mukhtar

**Affiliations:** 1Department of Biology, University of Alabama at Birmingham, 3100 East Science Hall, 902 14th Street South, Birmingham, AL 35294, USA; amajeed@clemson.edu (A.M.); jinb2112@uab.edu (J.L.); addiekni@uab.edu (A.J.K.); kmukhta@clemson.edu (K.M.P.-M.); 2Department of Genetics & Biochemistry, Biosystems Research Complex, Clemson University, 105 Collings St., Clemson, SC 29634, USA; 3Department of Biological Sciences, Clemson University, 132 Long Hall, Clemson, SC 29634, USA

**Keywords:** metagenome, 16S, coastal salt marsh, salt stress, plants

## Abstract

Salt marshes are highly dynamic and biologically diverse ecosystems that serve as natural habitats for numerous salt-tolerant plants (halophytes). We investigated the bacterial communities associated with the roots and leaves of plants growing in the coastal salt marshes of the Bayfront Beach, located in Mobile, Alabama, United States. We compared external (epiphytic) and internal (endophytic) communities of both leaf and root plant organs. Using 16S rDNA amplicon sequencing methods, we identified 10 bacterial phyla and 59 different amplicon sequence variants (ASVs) at the genus level. Bacterial strains belonging to the phyla *Proteobacteria*, *Bacteroidetes*, and *Firmicutes* were highly abundant in both leaf and root samples. At the genus level, sequences of the genus *Pseudomonas* were common across all four sample types, with the highest abundance found in the leaf endophytic community. Additionally, *Pantoea* was found to be dominant in leaf tissue compared to roots. Our study revealed that plant habitat (internal vs. external for leaves and roots) was a determinant of the bacterial community structure. Co-occurrence network analyses enabled us to discern the intricate characteristics of bacterial taxa. Our network analysis revealed varied levels of ASV complexity in the epiphytic networks of roots and leaves compared to the endophytic networks. Overall, this study advances our understanding of the intricate composition of the bacterial microbiota in habitats (epiphytic and endophytic) and organs (leaf and root) of coastal salt marsh plants and suggests that plants might recruit habitat- and organ-specific bacteria to enhance their tolerance to salt stress.

## 1. Introduction

Coastal salt marsh ecosystems are characterized by harsh environmental conditions, including periodic flooding, high salinity levels, varying herbivore densities, and sea level rise. In the United States (US), nearly 30% of the land contains salt marsh ecosystems, which are biologically significant in several ways [1]. They possess several biologically important properties, such as high rates of primary productivity, organic matter mineralization, and the recycling of essential nutrients [2]. The plants and their associated bacterial communities in this ecosystem have developed specialized adaptive mechanisms for salt stress over time, distinct from those of terrestrial or freshwater plants [3,4].

Plant-associated bacteria play a crucial role in promoting the growth and productivity of their host plants under both biotic and abiotic stress [5]. Salinity is a significant abiotic stressor that affects plant growth. It is known that plants in salt marsh ecosystems recruit beneficial microbes to enhance their resistance to salt stress [6]. Microbiota inhabiting the different spatial niches of these plants play important roles in shaping bacterial communities and enhancing host resistance to abiotic stress [7]. It is well established that salt-tolerant plants harbor specialized salt-tolerant microbiomes in their rhizosphere (roots) and phyllosphere (aerial parts of plants) [8]. These beneficial bacteria, known as salt-tolerant plant-growth-promoting bacteria (ST-PGPB), have been extensively studied to reveal their important roles in plant adaptation to salinity [9,10,11]. For instance, different strains of *Enterobacter* (endophytes), *Bacillus* (root endophytes), and *Pseudomonas* (the most abundant rhizobacteria) genera have been reported to induce salt tolerance and significantly improve the growth of Arabidopsis [12], rice [13], maize [14], and soybean [15]. Additionally, phyllosphere-associated bacterial communities are increasingly recognized as an important area of study under salt stress. Some studies have shown that increased rhizosphere salinity induces changes in the bacterial community structure and functions of the leaf phyllosphere, leading to reduced bacterial diversity and altered relative abundance [8,11].

Thus far, the bacterial community of the rhizosphere soil and root endophytes associated with salt marsh plants has been studied [16,17,18]. Studies focusing on both phyllosphere and rhizosphere surface (epiphytic) and within-plant (endophytic) communities are limited and provide restricted insights into the composition of bacterial communities in plants growing in coastal salt marsh ecosystems. Understanding how bacterial communities associated with marsh plants are influenced by their environment is critical for determining their ability to respond to environmental stressors. Here, we aimed to characterize and compare the assembly of bacterial epiphytes and endophytes associated with the phyllosphere (leaves) and rhizosphere (roots) of coastal salt marsh plants using 16S rDNA amplicon sequencing methods. The results of this study will help identify salt-tolerant plant organ- and habitat-specific bacterial communities within the salt marsh ecosystem.

## 2. Materials and Methods

### 2.1. Plant Sample Collection and Preservation

To investigate the microbial communities that support plant growth in sandy areas, we focused on the Bayfront Beach areas known for a high density of plant communities in sandy conditions. We first surveyed the prevalence of plant species in this area and selected the most predominant species. Two individual plants of the same species were collected from Bayfront Beach, Mobile, Alabama, on 15 October 2022. These plants had complete root and leaf systems and were in the flowering stage. Each plant was sampled individually into Ziploc bags and immediately transferred to an ice box for transportation to the laboratory. We performed three replicates for each plant sample collected.

### 2.2. DNA Sample Preparation

For epiphytic DNA isolation, plant tissues (roots or leaves) were collected into 2 mL centrifuge tubes and eluted in 1× Phosphate Buffered Saline (PBS) buffer for 10 min. The eluate was then processed using DNeasy PowerSoil Pro Kits (QIAGEN, Hilden, Germany, Catalog Number: 47014) with minor modifications. Briefly, 200 µL of eluate was mixed with 800 µL of Solution CD1 and vortexed briefly, followed by centrifugation at 15,000× *g* for 1 min to remove debris. Subsequent steps followed the manufacturer’s protocol. For endophytic DNA isolation, samples were initially washed with 70% ethanol for 40 s and then with 2% bleach for 1 min 20 s to eliminate all epiphytic DNA. The sterilized tissues were rinsed in sterile water for 5 times. Subsequently, approximately 300 mg of tissue was used for DNA isolation using the DNeasy PowerSoil Pro Kits without modification.

### 2.3. Library Construction and High-Throughput DNA Sequencing of 16S rDNA

The V4 regions of microbial 16S rDNA were amplified using primers targeting the 515F-Y (5′-GTGYCAGCMGCCGCGGTAA-3′) and 806R (5′-GGACTACNVGGGTWTCTAAT-3′). PCR amplifications were conducted using Phusion High-Fidelity DNA Polymerase (Thermo Scientific, Waltham, MA, USA, Catalog Number: F-530XL) according to the manufacturer’s instructions. The PCR cycling conditions consisted of 1 cycle of 95 °C for 5 min, followed by 30 cycles of 95 °C for 1 min, 55 °C for 1 min, and 72 °C for 30 s. A final extension step was performed at 72 °C for 5 min, followed by holding at 4 °C. PCR products were visualized on 1% agarose gels to confirm successful amplification results. Clean-up of the PCR products containing amplified 16S rDNA V4 regions was performed using the ExoSAP-IT PCR Product Cleanup kit (Affymetrix Inc., Santa Clara, CA, USA, Catalog Number: 78250.40.UL) following its protocol to remove excess dNTPs and primers. Subsequently, amplicon sequencing libraries were constructed using the NEBNext^®^ Ultra II DNA Library Prep Kit for Illumina^®^ platforms (New England Biolabs, Ipswich, MA, USA, Catalog Number: NEB #E7103). Final cDNA library quality was assessed using the Agilent 2100 Bioanalyzer (Agilent, Santa Clara, CA, USA). Libraries meeting quality criteria were subjected sequencing on the MiSeq System available at the Genomics Core Facility, University of Alabama at Birmingham.

### 2.4. Bioinformatics Analysis

The raw 16S rDNA was quality-filtered fastp (v.0.23.4) [19]. Read Quality filtering, denoising, and chimeric sequence removal were done using the Qiime2 DADA2 denoising method [20]. Before that, we removed low-quality reads with an average quality score below 30, valid primer sequence or barcode sequence, and ambiguous bases. The DADA2 pipeline clustered the sequences into an amplicon sequence variant (ASV) table, which provides a higher-resolution alternative to traditional OTU tables. A Naïve Bayes classifier trained on Greengenes 2022.10 99% OTUs 16S rRNA database was used to classify ASVs [21]. All features classified as “chloroplast”, “Archaea” and “mitochondria” were removed from the 16S rRNA data set.

### 2.5. Statistical Analysis

Statistical analyses were performed with the vegan package [22] in R (v.4.3.1). Paired Kruskal-Wallis test was used to test for differences in sample alpha diversities (observed OTUs and Shannon, Evenness indices). Taxabarplots and Venn diagram were generated using the R package “microeco” (v.1.7.1) [23]. The PCoA analysis based on Bray-Curtis and Jaccard distance was performed to determine and visualize dissimilarity in bacterial communities among plant organs (roots and leaves) and plant habitat (i.e., epiphytes and endophytes). The analysis of variance (ANOVA) was carried out using distance matrices (Adonis) with 999 permutations to determine significant differences in the bacterial community composition among habitats.

### 2.6. Microbial Network Construction and Analysis

In this study, corMicro (correlation matrices calculation for microbiome networks) function was used to construct microbial interaction networks. Phyllosphere and rhizosphere for epiphyte and endophyte interaction networks were constructed. A publicly available R package ggClusterNet [24] was used to construct and analyze the network. Only taxa detected in each set of more than two samples were retained in network construction. The Spearman coefficient was used to calculate the correlation.

## 3. Results

### 3.1. Microbial Community Composition and Diversity

We employed high-throughput sequencing to investigate bacterial diversity and composition across all samples. A total of 59 Amplicon Sequence Variants (ASVs) at the genus level were identified, spanning 10 phyla (Figure 1A). *Proteobacteria* (90.75%) emerged as the most abundant phylum across all samples; however, its relative abundance varied among different groups, accounting for 22.90%, 35.17%, 24.20% and 17.70% within leaf endophytes, leaf epiphytes, root endophytes, and root epiphytes, respectively. *Bacteroidota* (8.0%) was the second most dominant phylum, showing relatively high abundance in leaf and root epiphytes. *Firmicutes* (1.5%) were exclusively observed in leaf samples, whereas their presence in root samples was less than 1%. In addition to well-known bacterial groups, the plants were also colonized by newly classified or unknown ones, including the newly defined superphylum *Patescibacteria* [25].

The bacterial abundances at the genus level exhibited noticeable differences across samples (Figure 1B). *Pseudomonas*, *Pantoea*, *Rhizobium*, *Pedobacter*, *Duganella*, *Flavobacterium*, and *Bacillus* were dominant genera in the groups, though their abundances varied. Specifically, *Pantoea* was more prevalent in both the epiphytic and endophytic compartments of the leaf (12.2% and 22.6%, respectively) compared to the root (5.4%). *Rhizobium* showed lower abundance in leaf endophytes (6.2%). Additionally, *Duganella* exhibited a higher proportion (5.4%) in both leaf and root endophytic compartments. The relative abundance of *Pseudomonas* was notably higher in leaf endophytes (58.1%) than in the other samples. These findings suggest that these bacterial genera may play a crucial role in enhancing plant tolerance to salt stress.

### 3.2. Shared and Unique ASVs and Keystone Taxa

A total of 154 Amplicon Sequence Variants (ASVs) were detected across all libraries, with 35 ASVs common to all samples (Figure 1C). The numbers of ASVs exclusive to the leaf epiphyte and endophyte groups were 20 and 1, respectively, and for root endophyte and epiphyte were 6 and 5. Three ASVs were common between leaf endophytes and epiphytes, while 10 common ASVs were found between root endophyte and epiphyte groups. The shared ASVs mainly belonged to the *Proteobacteria*, *Bacteroidetes*, and *Firmicutes* phyla.

### 3.3. Microbial Co-Occurrence Network

Network analyses were conducted using bacterial ASVs to explore microbial interactions and differences between leaf and root endophytic and epiphytic communities. These analyses are crucial for deciphering co-occurrence patterns across microbial taxa within complex communities and investigating potential positive and negative interactions among diverse taxa. We constructed two sets of co-occurrence networks: endophytic and epiphytic. Within each category, networks were developed for both leaf and root samples. Each network exhibited distinct topological features (Figure 2). Generally, epiphytic networks contained a greater number of nodes and edges compared to endophytic networks. Conversely, among endophytic networks, those of the roots displayed the highest number of nodes and edges. These findings indicate varied levels of bacterial taxa complexity between endophytic and epiphytic networks, particularly in the roots. Notably, the endophytic network of root samples showed the highest number of negative edges, suggesting the presence of antagonistic interactions among diverse ASVs. Furthermore, our results indicated that a more diverse array of ASVs participated in the interactions within epiphytic networks compared to endophytes. *Proteobacteria*, *Bacteroidota*, *Acidobacteria*, *Actinobacteria*, *Bdellovibrionota*, *Firmicutes*, and *Patescibacteria* were predominantly detected in the endophytic networks (Figure 2A). Additionally, *Myxococcota*, *Verrucomicrobiota*, and *Planctomycetota* were exclusively detected in the epiphytic networks (Figure 2B).

Connectivity, or node degree, reflects the strength of connections between nodes in a network. By examining bacterial phyla with the highest connectivity, often considered keystone taxa in networks, we observed varying abundances of these keystone phyla across different groups. In the endophytic network, *Patescibacteria* and *Acidobacteriota* ranked among the top 10 nodes with high degrees in roots, but were absent from the list for leaves. Additionally, *Planctomycetota* appeared exclusively in the top 10 nodes with high degrees in the epiphytic root network. *Verrucomicrobia* and *Myxococcota* consistently ranked in the top 10 nodes with high degrees across all epiphytic networks. These specific microbial taxa played distinct roles in network structures for each group. These findings indicate varied levels of OTU complexity in endophytic and epiphytic networks of roots and leaves.

### 3.4. Bacterial Diversity Analyses Differentiate Communities among Habitats

For alpha diversity analysis, which assesses diversity within each sample, we calculated diversity, richness, and evenness indices (Figure 1E,F). Root epiphytic samples showed higher richness compared to leaf samples (*p* = 0.032). Evenness was lower in both leaf and root epiphytic communities compared to leaf and root endophytic communities (*p* = 0.05). Beta diversity analysis was conducted to examine the effects of ‘organ’ (Phyllosphere vs. Rhizosphere), ‘habitat’ (epiphyte vs. endophyte), and organ + habitat (leaf epiphytes vs leaf endophytes and root epiphytes vs root endophytes) on variations in species composition. The Adonis test results indicated a significant effect of ‘habitat’ (R2 = 0.17, *p*-value = 0.01) on microbial communities (Figure 1D).

## 4. Discussion

Salt marshes serve as natural habitats for numerous salt-tolerant plants [26], and the unique morphological, physiological, and molecular adaptations of these plants make them an interesting subject of study. However, our current understanding of their microbial community structure, specifically the phyllospheric (leaf) and rhizospheric (root) components and their interactions within endophytic and epiphytic habitats, remains limited.

In this study, we explored the bacterial community composition, focusing on the phyllospheric and rhizospheric components of plants growing in the salt marsh of Bayfront Beach, Mobile, USA. *Proteobacteria*, *Bacteroidetes*, and *Firmicutes* were identified as the most abundant phyla colonizing both the phyllosphere and rhizosphere (Figure 1A), while bacteria from the *Actinobacteria* phylum were less prevalent. These findings regarding the dominant phyla are consistent with earlier studies of phyllosphere and rhizosphere bacteria reported in various salt-tolerant plant species within salt marsh ecosystems [8,11,27]. For example, Szymańska, S., et al. [28] similarly found a high abundance of bacteria belonging to *Proteobacteria*, *Actinobacteria*, and *Firmicutes* phyla in the root-associated endosphere under salt stress. Qu, X., et al. [11] reported *Proteobacteria*, *Bacteriodetes*, and *Firmicutes* in the phyllosphere leaf community of *Tamarix chinensis*. All these studies have focused on endophytes of the root and phyllosphere.

*Proteobacteria* is a known common phylum associated with strong salt tolerance in plants [27]. In our study, the *γ-Proteobacteria* were more diverse compared to *α*- and *β-Proteobacteria*. It has been reported that salinization favored the dominance of *γ-Proteobacteria* [29]. Thus, resulting in a significant increase in rhizosphere *Proteobacteria* in a saline environment. A wide distribution of *γ-Proteobacteria* has been reported in sediments of marine wetland ecosystems, and most of them seemed involved in sulfur reduction [30]. The second most dominating bacterial phylum we found in our study was *Bacteroidota*, reported more often as the most characteristic phylum in salt marsh ecosystems [31,32]. Previous studies have reported that *Bacteroidetes* can assist in sulfur cycling and organic matter degradation. They can also donate electrons to sulfate-reducing bacteria, thereby promoting the growth of salt-tolerant plants [33]. The phyllospheric bacterial community exhibited a relatively high abundance of *Firmicutes* compared to the rhizosphere (Figure 1A). *Firmicutes*, along with *Proteobacteria*, *Bacteroidetes*, and *Actinobacteria*, demonstrate significant tolerance to a wide range of abiotic stresses, including salt stress typical of coastal environments, highlighting their adaptive capabilities to this habitat [34].

Our findings in this study revealed that the saline habitat (internal vs. external for leaves and roots) significantly influenced the bacterial community structure. This highlights the potential for different interactions between plants and their endophytes or epiphytes. Similar differences have been observed in plants growing in saline areas, such as *Avicennia marina* [8]. Additionally, differences in nutrients and environmental conditions between plant roots and leaves (epiphytes and endophytes) could have influenced the selection of specific bacterial OTUs, leading to distinct bacterial communities within epiphytes and endophytes. Notably, the genus *Pseudomonas* was found in all four sample types (Figure 1B), with its highest abundance observed in the leaf endophytic community (58% of sequences). Additionally, *Pantoea* sequences were less abundant in root samples (5% of sequences) compared to leaves (ranging from 12% to 22%). Many species of *Pantoea* have been extensively studied for their roles as growth promoters, biological control agents, and enhancers of broad-spectrum resistance against biotic and abiotic stresses [35]. Overall, the leaves and roots of the studied plant species are colonized by many of the same phyla and genera, although in varying proportions between endophytic and epiphytic groups. This suggests that many of the taxa found in both leaves and roots may originate from similar sources.

Network analysis can model the co-occurrence of microbial taxa, illustrating their interactions and clustering patterns. The network’s topological properties, including connectivity and modularity of individual taxa, can provide insights into their ecological roles within the network [36]. By studying the bacterial phyla with the highest connectivity, typically considered keystone taxa in a network, we found that the microbial network structure in the phyllosphere and rhizosphere responded differently in epiphytic and endophytic habitats. Our results indicated that a more diverse array of phyla participated in the interactions of root and leaf epiphytes compared to endophytes in bacterial networks. Generally, epiphytic networks exhibited a greater number of nodes, edges, and average node degrees compared to all endophytic networks. Conversely, among endophytic networks, the root plot displayed the highest number of nodes, edges, and average degree (Figure 2). These findings suggest varying levels of ASV complexity in epiphytic networks of roots and leaves.

## 5. Conclusions

This study contributes to elucidating the bacterial diversity and composition of coastal salt marsh plants using 16S rDNA sequencing, particularly in the context of salt stress. Dominant-associated bacteria included *Proteobacteria*, *Bacteroidetes*, and *Firmicutes*. We demonstrated that bacterial communities varied across different habitats (endophytes and epiphytes) of leaf and root plant tissues. Findings from our co-occurrence network analyses supported our understanding of microbial dynamics in response to salinity. Our results indicated that a more diverse array of phyla participated in interactions among root and leaf epiphytes compared to endophytes in bacterial co-occurrence networks. This study establishes a foundation for further research and underscores the importance of defining microbial diversity under salt stress conditions. Future studies are needed to elucidate the functional roles of these bacterial species in plant-microbe interactions within salt marsh ecosystems.

## Figures and Tables

**Figure 1 microorganisms-12-01595-f001:**
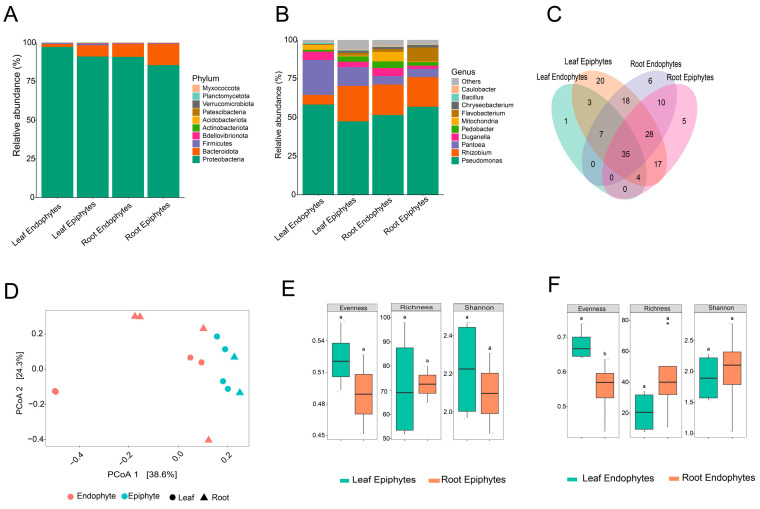
Bacterial community composition and diversity salt-marsh plants. (**A**,**B**) Group-wise relative abundance of dominating phylum and genus of root and leaf endophytes and epiphytes. (**C**) Shared and unique ASVs between all four groups. (**D**) Principal Coordinates Analysis (PCoA) of all samples based on Jaccard’s index (binary data). (**E**,**F**) An increase in microbial richness and diversity was found in roots compared to leaves in epiphytic habitats, revealed by Chao 1 and Shannon index.

**Figure 2 microorganisms-12-01595-f002:**
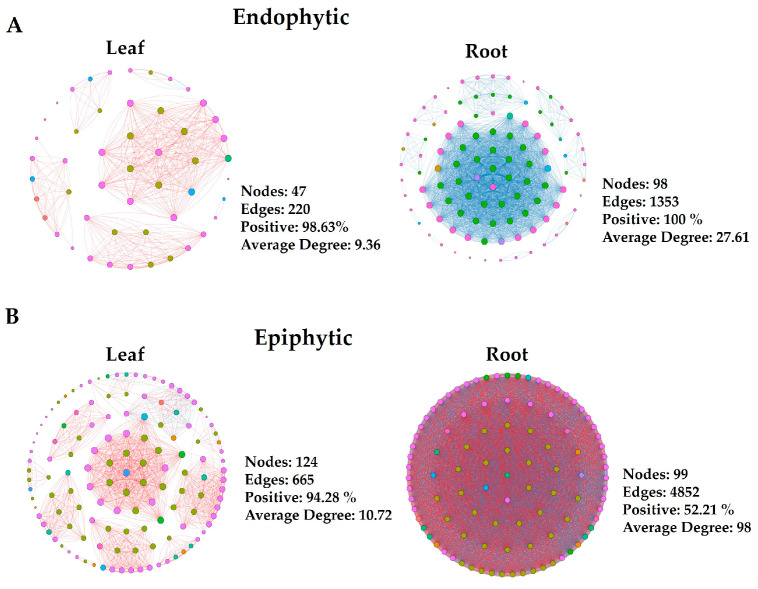
Four groups each for endophytic (**A**) and epiphytic networks (**B**) are illustrated. Individual types of networks within each category are indicated. With the node size proportional to node connectivity, the node color represents various phyla, while the line color indicates positive (red) and negative (blue) correlation coefficients. Network construction employed Spearman’s correlation coefficient, with r > 0.6 and *p* < 0.05 as criteria.

## Data Availability

The original contributions presented in the study are included in the article, further inquiries can be directed to the corresponding author.

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
