# Peer review of "Bacterial Communities Associated with the Leaves and the Roots of Salt Marsh Plants of Bayfront Beach, Mobile, Alabama, USA"

_microorganisms, 2024, doi:10.3390/microorganisms12081595_

Round 1

Reviewer 1 Report

Comments and Suggestions for Authors

The present research evaluates the bacteria communities associated to roots and leaves of plants of salt marsh, both endophytic and epiphytic.

The abstract is a good reflect of the work performed.

The introduction descrives the knowledge on the topic with adapted and correct references and states correctly the reseach question.

The material and method is well presented for the bacterial analysis but the main concern is that nothing is said about the plant where those bacteria were collected, we do not know if there were several species or just one that were harvested, at which stage of development, and at which date. They said that there were three replicates but without knowning which is inside except root/leaves parts. Then in the discussion they said that the litterature makes the analysis by plant specie, but here we do not know nothing about the vegetal material and if there are differences within species.

The results are clear (with the previous default) only figure 1 & 2 have not enough quality to be corrrectly readible.

The discussion is correct except the same previous default

Conclusions are ok

Minor other corrections in the document attached

Author Response

Comments 1: The present research evaluates the bacteria communities associated to roots and leaves of plants of salt marsh, both endophytic and epiphytic.

The abstract is a good reflect of the work performed.

The introduction descrives the knowledge on the topic with adapted and correct references and states correctly the reseach question.

  • The material and method is well presented for the bacterial analysis but the main concern is that nothing is said about the plant where those bacteria were collected, we do not know if there were several species or just one that were harvested, at which stage of development, and at which date. They said that there were three replicates but without knowning which is inside except root/leaves parts. Then in the discussion they said that the litterature makes the analysis by plant species, but here we do not know nothing about the vegetal material and if there are differences within species.

  • The results are clear (with the previous default) only figure 1 & 2 have not enough quality to be corrrectly readible.

The discussion is correct except the same previous default

Conclusions are ok

Minor other corrections in the document attached (see attached)

Response 1:  We have addressed the reviewer's comment by adding information about plant species and sample collection dates.

   On October 15, 2022 (fall season), two individual plants of the same species were collected from Bayfront Beach, Mobile, AL. These plants are native to the sandy beach area. To conduct a comprehensive investigation of both endophytes and epiphytes from different tissues, whole plants possessing complete root and leaf systems were collected. During bench work, the roots and leaves of both plants were separated, and bacterial isolation was conducted individually. We performed three replicates for each plant. Additionally, the plants were in the flowering stage at the time of collection. We could not identify the plant species exactly; based on our online searches and comparison using the photos we took; it could be Long Beach Primrose Willow or its relatives.

Based on the suggestions few modifications and insertions (old version in blue text and revised version in red text) have been made in the revised manuscript to strengthen its quality:

On page 2, lines 76-79 “Plant tissues were collected from beach areas of Bayfront Park, Mobile, Alabama, consisting of both aboveground (leaves) and underground (roots) parts. Three biological replicates for each plant species were sampled individually into Ziploc bags and immediately transferred to an ice box for transportation to the laboratory.” This has been modified to “To investigate the microbial communities that support plant growth in sandy areas, we focused on the Bayfront beach areas known for a high density of plant communities in sandy conditions. We first surveyed the prevalence of plant species in this area and selected the most predominant species. Two individual plants of the same species were collected from Bayfront Beach, Mobile, Alabama, on October 15, 2022. These plants had complete root and leaf systems and were in the flowering stage. Each plant was sampled individually into Ziploc bags and immediately transferred to an ice box for transportation to the laboratory. We performed three replicates for each plant sample collected.” Please check lines 75-82 on page 2.

As per the suggestion, the resolution for figures 1 & 2  has been increased to enhance its quality, and corrections have been made in the revised version of the manuscript. Please check on page 4 for figure 1 and page 6 for figure 2.

 On page 4 line 151, “Non-metric multidimensional scale (NMDS)” has been replaced withPrincipal Coordinates Analysis (PCoA) in the revised version of the manuscript. Please check the highlighted portion on line number 155 on page number 4.

On page 7 lines 255-257, In our study, plant habitat (internal vs. external for leaves and roots) was found to 255 be a determinant of the bacterial community structure, consistent with findings described 256 previously for other plant species [7, 8]. Differences” have been modified to “Our findings in this study revealed that the saline habitat (internal vs. external for leaves and roots) significantly influenced the bacterial community structure. This highlights the potential for different interactions between plants and their endophytes or epiphytes. Similar differences have been observed in plants growing in saline areas, such as Avicennia marina [8]. Additionally, differences. “ Please check lines 258-262 on page 7.

On page 7 line 266, “ studied Plants” has been modified to “ plant species” Please check line 271 on page 7.

Reviewer 2 Report

Comments and Suggestions for Authors

The paper performs an interesting descriptive study, but the contextualization of the study is very poor and this compromises the utility of the results. there is essential information that should be included before being able for acceptance.

a) Which is the binnomial name of  Daphne? It was a subspecie?

b) Why this plant was chosen? Is endemic to the area? Invasory? predominant? Which is its role in the ecosystem?

c) The sampling seems very limited (three samples from each tissue). But young plants? old plants? Young leaves? Primary or secondary roots? The plants where next to the sea or away? The microbiome may change depending on these factors. 

d) In which time of the year was the sampling performed? Can you provide the exact date? the microbiome may change depending on the season of the year or the physiological status or the plant (bolting? flowering? vegetative growth?)

e) How was the pluviometry before the sample collection. When plants siffer a drought the experience serious changes that may affect its related microbiome.

Please,provide all this information. 

Author Response

2nd Reviewer’s comment:

Comments 2: The paper performs an interesting descriptive study, but the contextualization of the study is very poor and this compromises the utility of the results. there is essential information that should be included before being able for acceptance.

  1. Which is the binomial name of Daphne? It was a subspecies.

  Response 2: Thank you for the questions. We aim to specify the collection site more accurately. The initial draft inaccurately described the location, so we have revised it to Bayfront Beach.

On page 1, line 3 “Daphne” This has been modified to “Bayfront Beach, Mobile,”

On page 1, line 16 “Daphne” This has been modified to “Bayfront Beach, Mobile,” Please check line 17 in revised version.

On page 6, line 229 “Daphne” This has been modified to “Bayfront Beach, Mobile,” Please check line 232 in revised version

  1. Why this plant was chosen? Is endemic to the area? Invasory? predominant? Which is its role in the ecosystem?

Response 3: The plant species were collected from the beach areas of Bayfront Parks on October 15, 2022. We first observed species growing on the beach and found them to be widespread, indicating they are endemic to this sandy environment. Additionally, these species grow isolated from other species. Considering the beach areas have high salt content, which inhibits the growth of most plant species, we collected these plants to isolate the microbiome community that might support their growth, such as by controlling water osmosis or reducing water evaporation.

  1. The sampling seems very limited (three samples from each tissue). But young plants? old plants? Young leaves? Primary or secondary roots? The plants where next to the sea or away? The microbiome may change depending on these factors. 

Response 4: These plants matured with fully developed leaves and complete root systems. We separated the leaves and primary roots for bacterial isolation. They were growing on the beach next to the sea, which is a very sandy area.

  1. In which time of the year was the sampling performed? Can you provide the exact date? the microbiome may change depending on the season of the year or the physiological status or the plant (bolting? flowering? vegetative growth?)

Response 5: We collected the plant species on October 15, 2022, during the fall season. Both individual plants within same species were collected in the flowering stage, with mature leaves and roots.

  1. How was the pluviometry before the sample collection. When plants siffer a drought the experience serious changes that may affect its related microbiome.

Response 6: We tracked the weather history of Bayfront Park, Mobile, AL, and found there was no precipitation before or during the collection period. The weather was sunny with an average temperature of 70.62°F. The plants did not experience significant weather changes before and during sample collection.

Other minor reviewer points:

The highlighted text on page 3 line 123 has been rephrased in the revised version of the manuscript. Please check the highlighted portion on line number 129 on page number 3.

Round 2

Reviewer 1 Report

Comments and Suggestions for Authors

The authors have now provided a corrected version of the document, where the main issue related to the description of the plant material has been well described. The document is ready to be published